# The Relationship between Mental Health and the Quality of Life of Polish Nurses with Many Years of Experience in the Profession: A Cross-Sectional Study

**DOI:** 10.3390/ijerph16101798

**Published:** 2019-05-21

**Authors:** Jolanta Lewko, Bianka Misiak, Regina Sierżantowicz

**Affiliations:** 1Department of Integrated Medical Care, Medical University of Bialystok, 15-089 Bialystok, Poland; 2Medical University of Bialystok Children’s Clinical Hospital, 15-274 Bialystok, Poland; bianka.misiak@o2.pl; 3Department of Surgical Nursing, Medical University of Bialystok, 15-089 Bialystok, Poland; renatasierz@wp.pl

**Keywords:** self-assessment, quality of life, satisfaction with life, mental health

## Abstract

Background: In their professional work, nurses struggle repeatedly with difficult situations that are causes of stress. Another issue is the low prestige of the nursing profession compared with other professions, which results in dissatisfaction, increased frustration, and lack of precision when performing professional tasks. The aim of this study was to assess the relationship between mental health and quality of life and satisfaction with life in nurses with many years of experience in the profession. Methods: The study was conducted in a group of 523 randomly selected professionally active nurses aged over 40 years old from the Podlaskie Voivodeship. Standardized questionnaires were used, including WHOQOL-BREF, a short version of a questionnaire assessing quality of life, the general health questionnaire (GHQ-28), and the satisfaction with life scale (SWLS). Results: The mental health component was found to be significantly affected by financial situation (*p* = 0.005). Among respondents describing their financial status as bad, the assessment of negative mental health symptoms was higher. The remaining studied variables—work experience, nature of work, place of residence, age, material status, having a partner, and having children—did not affect the respondents’ mental health status. The co-occurrence of chronic diseases affected (*p* = 0.008) the intensification of negative mental health symptoms such as somatic symptoms, anxiety, insomnia, and social dysfunction. The intensification of negative mental health symptoms was not connected with absence from work. Conclusions: The financial situation of the respondents significantly determined their quality of life as well as influencing mental health components. Nurses’ satisfaction with life was correlated with all studied domains of quality of life.

## 1. Introduction

A range of demographic and social factors affect the professional situation of nurses in Poland and throughout Europe. To analyze the professional situation of nurses in Poland, we used extensive information from the Supreme Council of Nurses and Midwives report 2015, which states that there were 283,066 nurses registered in Poland and 943 in the Podlaskie Voivodeship at the time of the report. Thus, the general employment rate of nurses per one thousand inhabitants was 4.82 for the whole country and 3.97 for the Podlaskie Voivodeship. The employment rate is projected to decrease; nowadays, the average employment rate of nurses in the Podlaskie Voivodeship is much lower than the national average [1].

The report also contains data on the above indicator for 16 selected countries. Poland is in last place, which indicates limited access to nursing care in this country. Switzerland has an indicator of 16, and not far below are Denmark and Norway with 15.4 and 14.4, respectively. Analyzing the age structure of the professional group of nurses, and thus the likelihood of retirement by 2022, 35,215 working nurses (28.67%) will leave the profession in Poland overall, including 1597 (33.91%) nurses in the Podlaskie Voivodeship [1]. The age of the nurses is problematic since most of them are around retirement age, and additionally, this professional group also has a low economic status [2]. An important fact is that, for financial reasons, nurses often work at two or more workplaces, which is associated with significant physical and mental burdens and stress.

In their professional work, nurses struggle repeatedly with difficult situations that are sources of stress. Prolonged stress causes many negative emotions, such as irritability and impatience. It also lowers motivation to work. Deteriorations in the relationships between nurses and patients and their relatives, and also among colleagues, can be observed. Another problem is the low prestige of the nursing profession compared with other professions, which results in disaffection, increased frustration, and lack of satisfaction with the performed professional tasks. A sense of lack of professional success and permanent fear can quickly lead to burnout, as well as reducing activity levels [3].

Nursing is a job with great responsibility while, at the same time, being one of the most difficult professions, because it is related to saving human health and life and often involves overcoming one’s own weaknesses [4,5]. The requirements and expectations of nurses have increased considerably. Therefore, it is necessary for nurses to be motivated and have a sense of satisfaction with their professional tasks as well as their general life situation so that they can meet these requirements and expectations better. Nursing can be a source of inspiration, but it can also be a huge burden [6,7,8]. This suggests the need to conduct research to determine the effects of age, quality of life, life satisfaction, and other work-related factors such as shift work, salary, and job satisfaction on the mental health of nurses. This is of great importance for the emotional well-being of nurses, which, in turn, can affect the quality of care provided to patients. The aim of the study was to assess the relationship between mental health, quality of life, and satisfaction with life in nurses with many years of experience in the profession.

## 2. Materials and Methods

### 2.1. Design and Participants

The study was conducted in a group of randomly selected professionally active nurses aged over 40 years old from the Podlaskie Voivodeship. The inclusion criteria were (1) individuals aged 40 years old or above and (2) individuals working as nurses. Nurses working in hospitals, various acute ward types, primary health care, outpatient specialist care, and others who agreed to participate were included in the study. The questionnaires were given in a sealed envelope to all nurses who were 40 years old and older in selected hospitals. A total of 560 questionnaires were distributed with the collaboration of the Chamber of Nurses and Midwives, and 523 envelopes were collected, and they were the basis for the empirical analysis.

The collected material has been encoded, saved in an Excel spreadsheet, and subjected to verification. Only completed questionnaires were included in the statistical analysis.

The study was based on a diagnostic survey using standardized psychometric questionnaires: The general health questionnaire (GHQ-28) [9], the quality of life WHOQOL-BREF [10], the satisfaction with life scale (SWLS) [11], and an ad-hoc questionnaire comprised of questions pertaining to the family and social situation, as well as information about the respondent, were included.

The general health questionnaire (GHQ-28) by David Goldberg was adapted into the Polish language by Makowska and Merecz, and the reliability of the test ranges from +0.82 to 0.93. The GHQ-28 is used as a screening test to reveal the presence of mental disorder and to enable the mental health of different populations to be studied. The questionnaire has a characteristic scale structure, which allows it to measure four health dimensions: scale A—somatic symptoms; scale B—anxiety and insomnia; scale C—social dysfunctions; and scale D—symptoms of depression. Each scale consists of 7 questions. The respondents themselves mark the answers on the Likert scale that most accurately reflect their feelings, which is typical for self-report questionnaires. The overall score on the GHQ 28 is the sum of the points that were obtained for all the questions in the questionnaire; thus, the score can be a maximum of 28 points. According to the recommendations of the questionnaire’s authors, apart from the overall score, it is possible to interpret the results obtained from the individual subscales [9,10].

The quality of life WHOQOL-BREF questionnaire consists of 26 questions that enable the assessment of the quality of life profile in four domains: physical, psychological, social functioning, and functioning in the environment. A respondent can receive up to 20 points in each domain—the higher the score, the better the quality of life. The calculations were done using the syntax WHOQOL-BREF program, and then the results were transformed into comparable results obtained with the full version of the quality of life WHOQOL-100, which enabled the results of different domains to be interpreted and compared using Cronbach’s alpha coefficients ranging from 0.92 to 0.94 [11,12]. The WHOQOL-BREF was adapted into the Polish language by Jaracz et al. [12].

The satisfaction with life scale (SWLS) by Diener, Emmons, Larsen, and Griffin was adapted into the Polish language by Juczyński. The questionnaire is used for individual and group examinations of healthy, as well as sick, adults. The SWLS result is a general indicator of satisfaction with life. In the American version, the results obtained correlate at a medium to high level with other measures of subjective well-being and selected personality characteristics. They correlate positively with self-esteem, but negatively with measures of neuroticism and emotionality. For the Polish version of this questionnaire, similar statistical values have also been obtained and the reliability indicator (Cronbach’s alpha) of the SWLS for adults is 0.81. The questionnaire consists of 5 statements, and respondents choose 1 of 7 possible answers. Respondents evaluate to what extent each of the statements describes to their life up to that point. Results range from 5 to 35 points—the higher the result, the greater the feeling of satisfaction with life [13]. 

### 2.2. Procedure and Ethical Considerations

The study was performed from January until December 2015. The research conformed with good clinical practice guidelines, and the followed procedures were in accordance with the Helsinki declaration. All nurses signed a consent form to participate in the study. The research was approved by the Bioethics committee of the Medical University of Bialystok (Resolution no. R-I-002/521/2014).

### 2.3. Statistical Analysis 

During the study, we created a database (a source of variables) with nurses’ statements, opinions, and assessments, enabling the use of computational techniques. Relationships between selected factors and psychometric measures were analyzed. Depending on the variables being compared, the Chi-square test for independence and Spearman’s rank correlation analysis were used. The Mann–Whitney test and the Kruskal–Wallis test were used to assess the significance of differences between the groups. *p* < 0.05 was regarded as statistically significant.

## 3. Results

### 3.1. Characteristics of the Studied Group

Among the studied nurses, the largest group included people aged 41–45 years old, and the next was those between 46 and 50. A total of 19% of the respondents were in the range of 51–55 years, and 12% were between 56 and 60. Two-thirds of the nurses had worked in the profession for over 20 years, and the vast majority (79%) did shift work. Every tenth studied nurse (11%) had a managerial position. Almost one-quarter of the respondents (23%) had a master’s degree, 36% had a bachelor’s degree, and 38% had secondary school level education. Half of the respondents were employed in one place, almost 30% had two places of employment, and every sixth person worked at more than two places. 

### 3.2. Factors Affecting Mental Health Status (GHQ-28)

The financial situation significantly determined the mental health components. Among respondents describing their financial status as bad, the assessment of negative mental health symptoms was higher. Detailed data are presented in Table 1. The remaining examined variables—place of residence (*p* = 0.262), work experience (*p* = 0.874), having children (*p* = 0.072), age (*p* = 0.125) (Table 1), doing shift work (*p* = 0.210), holding a managerial position (*p* = 0.838), and the number of places of employment (*p* = 0.318)—did not affect the respondents’ mental health status (Table 2). Marital status also did not have a statistically significant effect on the studied nurses’ mental health status (Table 1).

The most common chronic disease among the studied nurses was hypertension (11%), and more than 8% had spinal diseases (degeneration and discopathy). The co-occurrence of a chronic disease had a statistically significant effect on the severity of negative mental health symptoms, such as somatic symptoms, anxiety or insomnia, and social dysfunction. This effect was particularly clear for the general score; mental disorders were about 1.4 points higher (if mean values are compared) and 2.5 points higher (if median values are compared) in the group of people with a chronic disease. This correlation was statistically significant (*p* = 0.008). Detailed data are presented in Table 2.

The intensification of negative mental health symptoms was not connected with absence from work. To conduct the analysis, the nurses were divided into three separate groups: those who had no absences from work, those who had less than seven days of absence, and those who were absent for a week or more. Detailed data are presented in Table 2. 

### 3.3. Mental Health Status and Satisfaction with Life as well as Quality of Life

In this part of the study, mental health status, as measured by the GHQ, was taken into account, and its association with other selected psychometric measures was assessed.

When analyzing the descriptive measures of satisfaction with life using the SWLS, it was possible to obtain values in the range of 5–35 points, where higher values indicate greater satisfaction with life. The mean score achieved for the satisfaction with life scale in the study group was 21.5 ± 5.6 (median: 21). 

A summary of the descriptive statistics for the overall GHQ-28 results allowed us to conclude that the level of negative psychometric symptoms in the studied group was rather low. This was supported by the average values of individual measures, which were much closer to the minimum than the maximum.

The studied nurses had a relatively high quality of life (average value for each measure was approximately 55–65 points). The highest quality of life pertained to the social and psychological domains, while lower values (by a few points) were obtained for the other two domains. When analyzing the results, it is worth noting that even the lower quartile of the quality of life measures was at least 50 points, which means that at least three-quarters of the respondents assessed their quality of life at this level or higher (Table 3).

Next, correlations between quality of life and measures of mental functioning disorders calculated on the basis of the GHQ were examined. All obtained correlations were statistically significant and negatively correlated, which corresponds to the expectation that better quality of life translates into better mental health status and vice versa. Quality of life in the somatic and psychological domains was more strongly correlated with the level of mental health disorder than the other two domains. These correlations reached an absolute value of about 0.50. Similar conclusions were drawn by analyzing the impact of satisfaction with life on the occurrence and severity of mental health disorders. However, these correlations were weaker than for some of the WHOQOL-BREF domains. Data are provided in Table 4. 

## 4. Discussion

This study assessed the relationship between mental health and quality of life and satisfaction with the life of nurses with many years of experience in the profession. Our research analysis shows that the respondents’ financial situation, as was the case for their quality of life, significantly determined their mental health components. Respondents in a worse financial situation had higher assessment of negative mental health symptoms. Other sociodemographic variables, such as age, place of residence, work experience, having a partner, having children, and the nature of work, did not have a statistically significant impact on mental health status. 

Ibrahim et al. [14] reported that nurses’ low incomes or insufficient remuneration for work negatively affected their mental health and improving nurses’ incomes is necessary to improve their quality of life and job satisfaction. The study by Zdończyk and Rynkiewicz [15] revealed that the psychological domain of women’s lives depends, in a statistically significant way, on their financial situation. The authors noticed that functioning in the psychological domain improved with an upgrade of the financial situation. Andruszkiewicz et al. [16] showed that complaints about anxiety, depression, somatic symptoms, social functioning disorders, and sleep problems were more frequently reported by nurses who assessed their income and financial situation negatively. This was also confirmed by Kanadys et al. [17], who found that the occurrence of depressive symptoms was conditioned by the financial situation of women in the perimenopausal period. They occurred more often in women with a poor or average financial situation than in people who assessed their financial situation as good or very good. The main reasons for mental health problems among Chinese nurses, according to Tong et al. [18], are low professional status, unfavorable nature of work, including shift work, and a large amount of work, which cause depressive and somatic symptoms.

However, in a review of 37 articles (32 quantitative, four qualitative, and one mixed) by Tahghighi et al. [19], no definitive evidence was found that shift work is associated with the worse mental functioning of nurses.

Our research showed that shift work significantly affected symptoms of depression (GHQ-28) in the studied nurses, but this was not a statistically significant difference (*p* = 0.056).

Women aged 45–60 are at risk of developing psychosomatic diseases. A frantic pace of life and excessive responsibilities, with simultaneous hormonal changes, have negative impacts on women’s health, in both the physical and mental domains (Kliszcz et al.) [20]. Studies by Sansoni et al. [21] showed that the level of satisfaction of nurses in Italian hospitals is low. Their results revealed dissatisfaction with the professional requirements, organizational structure, and professional advancement. Wargo-Sugleris et al., [22] suggested that the environment and aging successfully are important areas that affect job satisfaction and delay of retirement in older nurses. Another study by Milosevic et al. [23] stated that the satisfactory ability to work significantly improved the quality of life of nurses in all domains.

Based on our results, it can be concluded that the co-occurrence of a chronic disease has a statistically significant effect on the severity of negative mental health symptoms, such as somatic symptoms, anxiety or insomnia, and social dysfunction. This effect was particularly visible for the general score. Simultaneously, the intensification of negative mental health symptoms was not found to be connected with absence from work or marital status. Quality of life in the somatic and psychological domains was more strongly correlated with the level of mental health disorder. 

However, Kliszcz et al. [20] showed that nurses are characterized by high levels of anxiety and, concerning those with the shortest work experience, aggression and depression were at low and average levels. 

Andruszkiewicz et al. [16] examined the impact of the work environment on the health of 364 nurses. The GHQ-28 allowed them to observe many significant correlations between the work environment and health status. In the studied group, the most common health problems were anxiety and sleep problems, as well as dissatisfaction within the physical state, while depressive type disorders were almost never revealed. Work experience and age correlated positively with the occurrence of somatic symptoms. Older nurses who had been in the profession for longer were more likely to have sleeping problems, anxiety, depression, somatic symptoms, and worse social functioning. 

According to Gallagher et al. [24], older nurses must also pay attention to the risk of cardiovascular disease and regularly monitor their blood pressure. Workplaces that support lifestyle behaviors that prevent the risk of hypertension have also been shown to improve nurses’ behaviors and reduce the impact of their health on the productivity of an aging workforce.

The results of Bazazan et al. [25] showed that mental health problems and fatigue are correlated with each other, and they supported the direct and indirect (through fatigue) impacts of quality of life on mental health problems in all WHOQOL-BREF domains. Maatouk et al. [26] conducted an intervention in small groups based on the theory of successful aging for nurses aged >45 years, which was effective at improving quality of life associated with mental health and other mental health-related outcomes. However, another study by Zdonczyk [27] reported that a higher educational level, high material status, and stable job condition were associated with higher indicators of healthy behaviors than a lower material status and lower educational level. In an analysis of research from Canada and the United States, Nowrouzi et al. [28] indicated areas related to the quality of life of nurses that require improvement, including educating new nursing graduates, opportunities for lifelong learning, promoting positive peer relations, stress reduction programs, and increased financial compensation. National nursing workforce studies are crucial to allow evidence-based policymaking to improve nursing human resources worldwide [29]. It is recommended that more psychological support for nurses is introduced, drawing attention to the stressful work environment and the risk of burnout syndrome, which can significantly affect the care and safety of patients. Also, greater support from the management in terms of financial matters and increased employment is needed, as these factors will translate into improved quality of care and satisfaction of the nurses, as well as preventing nurses from leaving the profession. 

A limitation of the study is the sample used. We suggest that this research should be expanded to cover all regions of Poland. Another limitation of the study could be the presence of a response bias, i.e., it cannot be concluded whether the nurses who refused to participate in the study had a better or worse quality of life, mental health status, and satisfaction with life or economic status than the nurses who participated in this study. 

## 5. Conclusions

The respondents’ financial situation significantly determined their quality of life and mental health status, and nurses with worse financial situations had more negative mental health symptoms. Nurses’ satisfaction with life was correlated with the level of quality of life in all studied domains. The co-occurrence of chronic disease intensified negative mental health symptoms.

## Figures and Tables

**Table 1 ijerph-16-01798-t001:** Selected socio-demographic factors and respondents’ mental health status.

Socio-Demographic Factors	GHQ-28 * Domains				
	Somatic Symptoms	Symptoms of Anxiety and Insomnia	Social Dysfunctions	Symptoms of Depression	GHQ-28 * Total
	Mean ± SD, Me	Mean ± SD, Me	Mean ± SD, Me	Mean ± SD, Me	Mean ± SD, Me
**Age**					
41–45, n = 203	2.0 ± 2.2, 1.0	2.2 ± 2.4, 1.0	0.9 ± 1.6, 0.0	0.4 ± 1.1, 0.0	5.6 ± 5.8, 4.0
46–50, n = 153	2.5 ± 2.2, 1.0	2.5 ± 2.4, 2.0	1.1 ± 1.8, 0.0	0.6 ± 1.4, 0.0	6.6 ± 6.2, 5.0
51–55, n = 97	2.2 ± 2.3, 2.0	2.5 ± 2.6, 2.0	1.1 ± 1.8, 0.0	0.6 ± 1.4, 0.0	6.4 ± 6.5, 5.0
over 55, n = 62	1.9 ± 2.4, 1.0	2.0 ± 2.6, 1.0	1.3 ± 2.4, 0.0	0.8 ± 1.8, 0.0	6.0 ± 8.2, 2.0
*p*-value	0.095	0.274	0.810	0.531	0.125
**Work experience (in years)**					
up to 15, n = 50	1.9 ± 2.2, 1.0	2.3 ± 2.6, 1.0	0.9 ± 0.9, 0.0	0.4 ± 0.9, 0.0	5.5 ± 5.6, 4.5
16–20, n = 128	2.2 ± 2.3, 2.0	2.2 ± 2.4, 1.5	1.1 ± 1.7, 0.0	0.6 ± 1.3, 0.0	6.1 ± 6.2, 4.5
21–25, n = 345	2.2 ± 2.3, 2.0	2.3 ± 2.5, 1.0	1.1 ± 1.9, 0.0	0.6 ± 1.4, 0.0	6.2 ± 6.6, 4.0
*p*-value	0.697	0.950	0.788	0.658	0.874
**Marital Status**					
single, n = 41	1.9 ± 2.1, 1.0	2.1 ± 2.6, 1.0	0.9 ± 1.8, 0.0	0.2 ± 0.7, 0.0	5.2 ± 6.0, 2.0
married, n = 396	2.2 ± 2.2, 2.0	2.4 ± 2.5, 2.0	1.1 ± 1.8, 0.0	0.6 ± 1.4, 0.0	6.3 ± 6.4, 5.0
widowed, n = 23	2.0 ± 2.7, 0.0	2.2 ± 2.9, 1.0	1.1 ± 2.1, 0.0	0.6 ± 1.6, 0.0	6.0 ± 2.0, 8.2
divorced, n = 63	2.1 ± 2.3, 1.0	2.1 ± 2.5, 1.0	0.9 ± 1.5, 0.0	0.4 ± 1.0, 0.0	5.5 ± 5.9, 4.0
*p*-value	0.693	0.701	0.875	0.465	0.471
**Financial situation**					
Good, n = 188	1.9 ± 2.2, 1.0	1.9 ± 2.3, 1.0	0.8 ± 0.9, 0.0	0.3 ± 0.9, 0.0	4.9 ± 5.4, 3.0
Average, n = 331	2.2 ± 2.3,1.0	2.4 ± 2.5, 2.0	1.2 ± 1.9, 0.0	0.6 ± 1.4, 0.0	6.3 ± 6.6, 5.0
Bad, n = 14	2.9 ± 2.4, 3.0	3.2 ± 2.8, 3.0	1.5 ± 2.0, 0.0	0.9 ± 1.6, 0.0	8.5 ± 7.2, 7.0
*p*-value	0.032	0.003	0.088	0.023	0.005
**Place of residence**					
big city 15, n = 241	2.2 ± 2.2, 2.0	2.2 ± 2.5, 1.0	1.0 ± 1.8, 0.0	0.5 ± 1.4, 0.0	5.9 ± 4.0, 6.5
small town 15, n = 221	2.1 ± 2.3, 1.0	2.3 ± 2.5, 2.0	1.1 ± 1.8, 0.0	0.5 ± 1.2, 0.0	6.0 ± 6.2, 4.0
village, n = 61	2.5 ± 2.4, 2.0	2.8 ± 2.5, 2.0	1.2 ± 2.0, 0.0	0.8 ± 1.4, 0.0	7.3 ± 6.8, 6.0
*p*-value	0.487	0.107	0.371	0.060	0.262
**Have children?**					
Yes, n = 470	2.2 ± 2.3, 2.0	2.4 ± 2.5, 2.0	1.1 ± 1.8, 0.0	0.6 ± 1.4, 0.0	6.2 ± 6.4, 5.0
No, n = 53	1.6 ± 2.0, 1.0	1.7 ± 2.2, 1.0	0.4 ± 1.2, 0.0	0.4 ± 1.2, 0.0	4.7 ± 6.0, 2.0
*p*-value	0.0527	0.0904	0.9851	0.1047	0.0725

* GHQ-28—general health questionnaire-28.

**Table 2 ijerph-16-01798-t002:** Selected professional and health factors and respondents’ mental health status.

Professional and Health Factors	GHQ-28 * Domains				
	Somatic Symptoms	Symptoms of Anxiety and Insomnia	Social Dysfunctions	Symptoms of Depression	GHQ-28 * Total
	Mean ± SD, Me	Mean ± SD, Me	Mean ± SD, Me	Mean ± SD, Me	Mean ± SD, Me
**Shift work**					
Yes, n = 413	2.2 ± 2.3, 2.0	2.3 ± 2.5, 2.0	1.1 ± 1.9, 0.0	0.6 ± 1.4, 0.0	6.3 ± 6.6, 5.0
No, n = 110	2.0 ± 2.2, 1.5	2.1 ± 2.5, 1.0	0.8 ± 1.5, 0.0	0.3 ± 0.9, 0.0	5.3 ± 5.5, 3.0
*p*-value	0.424	0.202	0.161	0.056	0.210
**Work in a managerial position**					
Yes, n = 56	2.2 ± 2.6, 1.0	2.2 ± 2.6, 1.0	1.1 ± 1.9, 0.0	0.6 ± 1.7, 0.0	6.1 ± 6.9, 4.0
No, n = 467	2.2 ± 2.2, 2.0	2.3 ± 2.5, 1.0	1.1 ± 1.8, 0.0	0.5 ± 1.3, 0.0	6.1 ± 6.4 ,4.0
*p*-value	0.899	0.542	0.712	0.663	0.838
**Number of jobs**					
One, n = 289	2.2 ± 2.2, 2.0	2.3 ± 2.5, 1.0	1.0 ± 1.7, 0.0	0.5 ± 1.2, 0.0	6.0 ± 6.0, 5.0
Two, n = 148	1.9 ± 2.2, 1.0	2.1 ± 2.4, 1.0	1.1 ± 2.0, 0.0	0.6 ± 1.6, 0.0	5.7 ± 6.9, 3.0
More, n = 86	2.6 ± 2.5, 2.0	2.5 ± 2.6, 2.0	1.2 ± 2.1, 0.0	0.7 ± 1.4, 0.0	7.0 ± 7.0, 4.0
*p*-value	0.148	0.511	0.710	0.290	0.318
**Absence from work**					
never, n = 367	2.1 ± 2.1, 1.0	2.3 ± 2.5, 1.0	1.0 ± 1.7, 0.0	0.5 ± 1.1, 0.0	5.8 ± 5.9, 5.0
less than a week, n = 55	2.4 ± 2.4, 2.0	2.1 ± 2.3, 1.0	1.1 ± 2.0, 0.0	0.8 ± 1.8, 0.0	6.4 ± 6.9, 4.0
week or more, n = 101	2.3 ± 2.5, 2.0	2.5 ± 2.7, 2.0	1.3 ± 2.1, 0.0	0.7 ± 1.7, 0.0	6.8 ± 7.8, 4.0
*p*-value	0.735	0.789	0.779	0.651	0.929
**Chronic comorbidity**					
No, n = 355	2.2 ± 2.2, 1.0	2.2 ± 2.5, 1.0	1.0 ± 1.7, 0.0	0.5 ± 1.3, 0.0	5.6 ± 6.2, 3.0
Yes, n = 168	2.5 ± 2.4, 2.0	2.6 ± 2.5, 2.0	1.3 ± 2.0, 0.0	0.6 ± 1.4, 0.0	7.0 ± 6.7, 5.5
*p*-value	0.019	0.038	0.033	0.479	0.008

* GHQ-28—general health questionnaire-28.

**Table 3 ijerph-16-01798-t003:** Average scores for the SWLS, GHQ-28, and WHOQOL-BREF among respondents.

Average Scores of Questionnaires	Mean	Me	*SD*	*c* _25_	*c* _75_	Min.	Max.
**SWLS *** CI 95% (20.2–21.9)	21.5	21	5.6	18	25	6	35
**GHQ-28 ***							
Somatic symptoms CI 95% (1.9–2.5)	2.2	2	2.3	0	4	0	7
Symptoms of anxiety and insomnia CI 95% (2.1–2.9)	2.3	1	2.5	0	4	0	7
Social dysfunction CI 95% (0.9–1.4)	1.1	0	1.8	0	1	0	7
Symptoms of depression CI 95% (0.3–0.8)	0.5	0	1.3	0	0	0	7
GHQ (Total) CI 95% (5.8–6.3)	6.1	4	6.4	1	10	0	28
**WHOQoL-Bref ***							
Somatic domain CI 95% (12.7–13.1)	12.9	13.1	1.9	12.0	14.3	4.6	17.7
Psychological domain CI 95% (13.5–13.9)	13.7	14.0	1.9	12.7	14.7	5.3	14.7
Social domain CI 95% (14.3–14.8)	14.6	14.7	2.9	13.3	16.0	4.0	20.0
Environment CI 95% (13.0–13.4)	13.2	13.5	2.2	12.0	14.5	5.0	19.0

* SWLS—satisfaction with life scale, GHQ-28—general health questionnaire-28, WHOQOL-BREF—quality of life, CI 95%—confidence interval.

**Table 4 ijerph-16-01798-t004:** Spearman’s rank correlation coefficients between quality of life and satisfaction with life and respondents’ mental health status.

Quality of Life WHOQOL-BREF *	GHQ-28 * Domains				
	Somatic Symptoms	Symptoms of Anxiety and Insomnia	Social Dysfunction	Symptoms of Depression	Total
Somatic domain	−0.45 *p* = 0.0000	−0.44 *p* = 0.0000	−0.32 *p* = 0.0000	−0.33 *p* = 0.0000	−0.50 *p* = 0.0000
Psychological domain	−0.36 *p* = 0.0000	−0.42 *p* = 0.0000	−0.39 *p* = 0.0000	−0.40 *p* = 0.0000	−0.47 *p* = 0.0000
Social domain	−0.19 *p* = 0.0000	−0.28 *p* = 0.0000	−0.25 *p* = 0.0000	−0.27 *p* = 0.0000	−0.28 *p* = 0.0000
Environment	−0.26 *p* = 0.0000	−0.28 *p* = 0.0000	−0.26 *p* = 0.0000	−0.20 *p* = 0.0000	−0.31 *p* = 0.0000
SWLS *	−0.23 *p* = 0.0000	−0.28 *p* = 0.0000	−0.21 *p* = 0.0000	−0.22 *p* = 0.0000	−0.30 *p* = 0.0000

* GHQ-28—general health questionnaire-28, WHOQOL-BREF—quality of life, SWLS—satisfaction with life scale.

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
