# Peer review of "The Relationship between Mental Health and the Quality of Life of Polish Nurses with Many Years of Experience in the Profession: A Cross-Sectional Study"

_ijerph, 2019, doi:10.3390/ijerph16101798_

Round 1

Reviewer 1 Report

I realize that a great work and time has been devoted to this paper. It is about the relationship between mental health and the quality of life of Polish nurses over 40 years old.  This is a topic of great significance to emotional wellbeing of nurses that can affect the quality of care provided to their patients, and therefore, be a public health problem. So I appreciate authors examining this topic.

The paper has a lot of strengths but I think that some changes should be recommended.

Abstract:

Readers should be able to read the abstract in isolation and understand what you have done, and its implications. It is not necessary to cite the authors of the scales used in the abstract. Please, avoid it.

Introduction:

The introduction is very scarce, is disorganized, severe lacking of references and does not justify why it has conducted this study.

Methodology:

It is unclear why do you have selected only nurses over 40 years old. If you want to analyze how affects the professional experience, it would be better to limit it by job seniority in the workplace. Age can be a moderator variable to be considered and it may be a bias in the results.

In line 74 when you talk about “original questionnaire comprised of questions”, I would suggest to write “ad-hoc” instead of “original”.

Could you specify how were conducted the questionnaires (e.g. e-mail, postal mail, at the workplace,)? And please, specify how long it takes to nurses performing all surveys.

Please, in the second paragraph of Materials and Methods, cite the scales correct in the text with the appropriate reference number, avoiding author’s names.

In line 95, you must correct the author’s names who are not properly cited.

Could you specify in the text the Cronbach’s Alpha Coefficient for the rest of the scales? Not only for the SWLS.

Results:

It is not clear why you have divided the sample into age groups.

The tables have not the last two columns in bold. I would like to know if it is for some special reason.

Please write the “p” always in lowercase, as in the last row of Table 2.

The +/- symbols look very close together in the tables.

Besides, please include a footnote in the tables with the definition of the abbreviations used.

The two paragraphs that begin in line 157 and 161 are not clear.

Discussion:

Please, unify the citations with point and coma (e.g. Finsterer et al.,) and the number of the reference must be next of the name of the author (e.g. Finsterer et al., [12]).

Please, avoid using abbreviations if you do not cite first in the text as “QoL” in line 248.

In line 256 when you talk about “group”, I suppose that the authors mean “sample”.

Conclusions:

Please, avoid the number 55 before “Conclusions”.

Your study has a lot of implications for nurses, hospital managements and even the patients. Please, can you elaborate some implications for practice and strategies to improve this situation? 

References:

Bibliographical references do not conform to the rules of the journal.

Please, read the author guidelines.

I hope that these recommendations do not discourage the authors and I want to recommend the authors to continue working on this paper, because it could be publishable if major changes are made.

Author Response

Comments and Suggestions for Authors

I realize that a great work and time has been devoted to this paper. It is about the relationship between mental health and the quality of life of Polish nurses over 40 years old.  This is a topic of great significance to emotional wellbeing of nurses that can affect the quality of care provided to their patients, and therefore, be a public health problem. So I appreciate authors examining this topic.

The paper has a lot of strengths but I think that some changes should be recommended.

Abstract:

Readers should be able to read the abstract in isolation and understand what you have done, and its implications. It is not necessary to cite the authors of the scales used in the abstract. Please, avoid it.

Response: The name authors of the scales were deleted.

Introduction:

The introduction is very scarce, is disorganized, severe lacking of references and does not justify why it has conducted this study.

Response: The literature has been supplemented by publication published in international journals.

Slusarska, B.; Zarzycka, D.; Dobrowolska, B.; Marcinowicz, L.; Nowicki, G.  Nursing education in Poland – The past and new development perspectives.   Nurse Educa Pract, 2018,31,118-125.

 The age of the nurses is problematic since most of them are around the retirement age and additionally, this professional group also has a low economic status.

 This suggests the need to conduct research determining the effect of age, quality of life, life satisfaction and other work-related factors such as shift work, salary, job satisfaction for the mental health of the nurses.

This is of great importance for the emotional well-being of nurses, which can affect the quality of care provided to patients.

Methodology:

It is unclear why do you have selected only nurses over 40 years old. If you want to analyze how affects the professional experience, it would be better to limit it by job seniority in the workplace. Age can be a moderator variable to be considered and it may be a bias in the results.

Response: As this is the largest group of working nurses, currently the average age of nurses in Poland is 49 years-oldTherefore, it was decided to carry out the research, especially in this age group.

In line 74 when you talk about “original questionnaire comprised of questions”, I would suggest to write “ad-hoc” instead of “original”.

Corrected

Could you specify how were conducted the questionnaires (e.g. e-mail, postal mail, at the workplace,)? And please, specify how long it takes to nurses performing all surveys.

Response: The study was conducted in a group of randomly selected, professionally active nurses aged over 40 years old from the Podlaskie Voivodeship. The inclusion criterion was age 40 years old or above and work as a nurse. Nurses working in hospital, various types of acute ward, primary health care, outpatient specialist care and other who agreed to participate were included in the study. The questionnaires were given in a sealed envelope to all the nurses who were 40 years old and older in a selected hospitals. A total of 560 questionnaires were distributed with the collaboration of the Chamber of Nurses and Midwives and in return 523 envelopes were collected and they were the basis for empirical analysis.

Please, in the second paragraph of Materials and Methods, cite the scales correct in the text with the appropriate reference number, avoiding author’s names.

Corrected

In line 95, you must correct the author’s names who are not properly cited.

Corrected

Could you specify in the text the Cronbach’s Alpha Coefficient for the rest of the scales? Not only for the SWLS.

Corrected

Results:

It is not clear why you have divided the sample into age groups.

Response: The authors woud like to check whether the age  determines the functioning of the mental health in the study grup of nurses.

The tables have not the last two columns in bold. I would like to know if it is for some special reason.

Editorial error

Please write the “p” always in lowercase, as in the last row of Table 2.

Corrected

The +/- symbols look very close together in the tables.

Corrected

Besides, please include a footnote in the tables with the definition of the abbreviations used.

Corrected

The two paragraphs that begin in line 157 and 161 are not clear.

Corrected

Discussion:

Please, unify the citations with point and coma (e.g. Finsterer et al.,) and the number of the reference must be next of the name of the author (e.g. Finsterer et al., [12]).

Corrected

Please, avoid using abbreviations if you do not cite first in the text as “QoL” in line 248.

Corrected

In line 256 when you talk about “group”, I suppose that the authors mean “sample”.

Corrected

Conclusions:

Please, avoid the number 55 before “Conclusions”.

Corrected

Your study has a lot of implications for nurses, hospital managements and even the patients. Please, can you elaborate some implications for practice and strategies to improve this situation?  

Response: It is recommended that this study introduces more psychological support into the practice of nurses, attention to the stressful work environment and the risk of burnout syndrome, as well as more support from the management in financial matters.

References:

Bibliographical references do not conform to the rules of the journal.

Please, read the author guidelines.

Corrected

I hope that these recommendations do not discourage the authors and I want to recommend the authors to continue working on this paper, because it could be publishable if major changes are made.

Thank you for your useful recommendation and support which will affect the quality of the manuscript.

Reviewer 2 Report

1)     I was not able to find several articles from the bibliography, for example there are no journals titled “public health and governance” or “nursing topics”. Some may be Polish publications, however, if ways to access these articles correctly can be provided, that would be helpful.

Reference #3 might be in Polish, “Cisek M, Przewoźniak L, Kózka M, Brzostek T, Brzyski P, Ogarek M, Gabryś T, Gajda K, 282 Ksykiewicz D.A: Workload during the last shift in the opinion of hospital nurses involved in 283 RN4CAST project. Public Health and Governance 2013; 11(2): 210-224”.I was able to find a related study by the same group, which I found interesting:

Brzostek T, Brzyski P, Kozka M et al:  Research lessons from implementing a national nursing workforce study. Int Nurs Rev. 2015 Sep;62(3):412-20.

2)     Search for the article “Zdończyk S.A., Rynkiewicz M: Quality of life of women after surgical treatment of breast cancer. 307 Polish Nursing 2015; 2(56): 153-158” produced the following:

Wpływ wybranych czynników socjomedycznych na jakość zycia i funkcjonowanie psychoseksualne kobiet po leczeniu operacyjnym raka gruczołu piersiowego. [The effect of selected socio-medical factors on quality of life and psychosexual functioning in women after surgical treatment of breast cancer]. Zdończyk SA. Pomeranian J Life Sci. 2015;61(2):199-206.

3)     This reference indicates that there has been improvements made in the education opportunities for nursing, with option to pursue bachelor’s, master’s and doctorate level training, which would open up greater opportunities.

Slusarska B, Zarzycka, D, Dobrowolska et al: Nursing education in Poland – The past and new development perspectives.  Nurse Education in Practice, 2018;31:118-125

4)     The finding that financial situation had a great impact on mental health, more so than other factors was interesting.  In the study, 23% of the respondents had a master’s degree, while 36% had a bachelor's – did the higher education not ensure higher salary, or were they in greater debt due to investment in higher education? If this significant finding could be elaborated more, in terms of other elicited information from the study, it would be helpful.

Author Response

1)    I was not able to find several articles from the bibliography, for example there are no journals titled “public health and governance” or “nursing topics”. Some may be Polish publications, however, if ways to access these articles correctly can be provided, that would be helpful.

Response: Public Health and Governance and Nursing Topics are the English name of the journal, we changed it to Polish name it's easier to find

Reference #3 might be in Polish, “Cisek M, Przewoźniak L, Kózka M, Brzostek T, Brzyski P, Ogarek M, Gabryś T, Gajda K, 282 Ksykiewicz D.A: Workload during the last shift in the opinion of hospital nurses involved in 283 RN4CAST project. Public Health and Governance 2013; 11(2): 210-224”.I was able to find a related study by the same group, which I found interesting:

Brzostek T, Brzyski P, Kozka M et al:  Research lessons from implementing a national nursing workforce study. Int Nurs Rev. 2015 Sep;62(3):412-20.

Thank you, we cited this study.

2)     Search for the article “Zdończyk S.A., Rynkiewicz M: Quality of life of women after surgical treatment of breast cancer. 307 Polish Nursing 2015; 2(56): 153-158” produced the following:

Wpływ wybranych czynników socjomedycznych na jakość zycia i funkcjonowanie psychoseksualne kobiet po leczeniu operacyjnym raka gruczołu piersiowego. [The effect of selected socio-medical factors on quality of life and psychosexual functioning in women after surgical treatment of breast cancer]. Zdończyk SA. Pomeranian J Life Sci. 2015;61(2):199-206.

Thank you, we cited this study.

3)     This reference indicates that there has been improvements made in the education opportunities for nursing, with option to pursue bachelor’s, master’s and doctorate level training, which would open up greater opportunities.

Slusarska B, Zarzycka, D, Dobrowolska et al: Nursing education in Poland – The past and new development perspectives.  Nurse Education in Practice, 2018;31:118-125

Response: In Poland, the age of the nurses is problematic since most of them are around the retirement age and this professional group also has a low economic status.

Thank you, we cited this study.

4)     The finding that financial situation had a great impact on mental health, more so than other factors was interesting.  In the study, 23% of the respondents had a master’s degree, while 36% had a bachelor's – did the higher education not ensure higher salary, or were they in greater debt due to investment in higher education? If this significant finding could be elaborated more, in terms of other elicited information from the study, it would be helpful.

Response: In Poland, higher education does not provide a higher salary, many nurses graduate from master's studies, which are not very expensive.

Thank you for your useful recommendation and support which will affect the quality of the manuscript.

Reviewer 3 Report

The article takes a very important topic both in Poland and in the world. Shortage of nurses and too much workload is a problem that affects many countries.

Major comments to address:

Point 1: The level of scientific writing is not at the standards expected for publication in a peer reviewed journal. Additionally, there are many grammatical errors throughout the paper. Closer review of the grammar and overall flow of the paper is needed.

Point 2: The introduction to this paper takes up the subject of the demographic situation among Polish nurses. In fact it is an analysis of the cited Report. No background presentation of the article, which would be related to the title: the quality of life and satisfaction with life, and why it is important for the nurse profession. Cited publications in the introduction (only Polish authors) point to a too shallow introduction to the topic.

Point 3: The purpose of the study is very vague, it may be due to language problems. Please present the purpose of the study in a more clear and understandable way so that it is consistent with the title of this paper.

Point 4: In the methodology section please describe in details how nurses were recruited and the respondents' workplace (hospital, type of department, primary health care, outpatient specialist care).

In addition, inclusion and exclusion criteria are missing.

More detailed information about the questions in the original questionnaire is required. How were these questions coded for analyses?

Minor comments to address:

Point 1: The Discussion section focuses on many aspects. It should be re-written to focus on only discussing the results found in this study as well as the implications of these findings.

Point 2: The limitations section should be broadened to include more specific limitations of this study beyond the generalizability.

Point 3: Tables clearly outline the results. You can only change the order: first show the level of life satisfaction, quality of life and general health among the examined nurses and then present correlations with the following factors.

Point 4: I have not found information about the data availability of this study. It should be indicated.

To sum up: Substantive and linguistic proofreading of the article is needed.

Author Response

The article takes a very important topic both in Poland and in the world. Shortage of nurses and too much workload is a problem that affects many countries.

 Major comments to address:

Point 1: The level of scientific writing is not at the standards expected for publication in a peer reviewed journal. Additionally, there are many grammatical errors throughout the paper. Closer review of the grammar and overall flow of the paper is needed.

Response: Grammatical and linguistic corrections were made.

The literature has been supplemented by publications published in international journals.

Slusarska, B.; Zarzycka, D.; Dobrowolska, B.; Marcinowicz, L.; Nowicki, G. 

Nursing education in Poland – The past and new development perspectives. 

Nurse Educa Pract, 2018,31,118-125.

Brzostek T, Brzyski P, Kózka M, Squires A, Przewoźniak L, Cisek M,  Gajda K,Gabryś T, Ogarek M. Research lessons from implementing a  national nursing workforce study. Int Nurs Rev. 2015 Sep;62(3):412-20.

Zdończyk SA. The effect of selected socio -medical factors on quality of life and 

psychosexual functioning in women after surgical treatment of breast cancer. 

Pom J Life Sci 2015, 61, 2, 199–206.

Point 2: The introduction to this paper takes up the subject of the demographic situation among Polish nurses. In fact it is an analysis of the cited Report. No background presentation of the article, which would be related to the title: the quality of life and satisfaction with life, and why it is important for the nurse profession. Cited publications in the introduction (only Polish authors) point to a too shallow introduction to the topic.

Response: The report presents the current demographic situation of nursing in  Poland.  The autors  focused on the largest group of nurses, i. e. people over 40 years of age in order to learn about their health condition and the factors that influence it.

Point 3: The purpose of the study is very vague, it may be due to language problems. Please present the purpose of the study in a more clear and understandable way so that it is consistent with the title of this paper.

Response: Changed to: The aim of the study was to assess relationship between mental health, the quality of life and satisfaction with life of nurses with many years of experience in the profession.

Point 4: In the methodology section please describe in details how nurses were recruited and the respondents' workplace (hospital, type of department, primary health care, outpatient specialist care).

Response: Changed to: The inclusion criterion was age 40 years old or above and work as a nurse. Nurses working in hospital, various types of acute ward, primary health care, outpatient specialist care and other who agreed to participate were included in the study. The questionnaires were given in a sealed envelope to all the nurses who were 40 years old and older in a selected hospitals. A total of 560 questionnaires were distributed with the collaboration of the Chamber of Nurses and Midwives and in return 523 envelopes were collected and they were the basis for empirical analysis.

In addition, inclusion and exclusion criteria are missing.

Response: Corrected: The inclusion criterion was age 40 years old or above and work as a nurse.

More detailed information about the questions in the original questionnaire is required. How were these questions coded for analyses?

Response: Corrected: The collected material has been encoded, saved in an Excel spreadsheet and subjected to verification. Only completed questionnaires were included in the statistical analysis.

Minor comments to address:

Point 1: The Discussion section focuses on many aspects. It should be re-written to focus on only discussing the results found in this study as well as the implications of these findings.

Response: We changed this part of our manuscript. It was very challenging but we made some changes.

Point 2: The limitations section should be broadened to include more specific limitations of this study beyond the generalizability.

Corrected

Point 3: Tables clearly outline the results. You can only change the order: first show the level of life satisfaction, quality of life and general health among the examined nurses and then present correlations with the following factors.

Response: Mental health status and satisfaction with life as well as quality of life are presented in table 3, correlation coefficient between quality of life and satisfaction with life and respondents' mental health status are presented in table 4.

Point 4: I have not found information about the data availability of this study. It should be indicated.

Response: Availability of data and materials

Please contact the corresponding author to discuss availability of data and materials.

To sum up: Substantive and linguistic proofreading of the article is needed.

Grammatical and linguistic corrections were made.

Thank you for your useful recommendation and support which will affect the quality of the manuscript.

Reviewer 4 Report

Thank you for submitting your manuscript on mental health and quality of life among Polish nurses over 40. You address an important issue in the profession, addressing factors that impact quality of life, life satisfaction,and mental health among older, more experienced Polish nurses. The use of standardized measures to measure the constructs under investigation, the use of random selection, and the excellent response rate on the questionnaires were definitely a strength of the study. In some cases, reliability data is reported on the instruments. I was unable to find any reference to reliability or validity data on the quality of life instrument. Reliability data is reported on the satisfaction with life scale. I was unable to locate data reported on the validity of the instruments. Also, could you clarify how questionnaires were distributed? Were they mailed to participants or handed out in person? Finally, I believe it would be helpful to the reader if you could expand your discussion on recommendations, particularly as they relate to practice. For example, are there resources that could be made available to nurses in the workplace to help them with mental health issues, financial concerns, etc.?

Author Response

Thank you for submitting your manuscript on mental health and quality of life among Polish nurses over 40. You address an important issue in the profession, addressing factors that impact quality of life, life satisfaction,and mental health among older, more experienced Polish nurses. The use of standardized measures to measure the constructs under investigation, the use of random selection, and the excellent response rate on the questionnaires were definitely a strength of the study.

In some cases, reliability data is reported on the instruments. I was unable to find any reference to reliability or validity data on the quality of life instrument. Reliability data is reported on the satisfaction with life scale. I was unable to locate data reported on the validity of the instruments.

Response: Changed to: The Quality of Life WHOQOL-Bref questionnaire consists of 26 questions and enables to obtain a quality of life profile in four domains: physical, psychological, social functioning, and functioning in the environment. A respondent could get up to 20 points in each domain; the higher the score, the better the quality of life. The calculations were done using the Syntax WHOQOL-Bref program, then the results were transformed to comparable the results obtained with the full version of the Quality of Life WHOQOL-100, which enabled interpreting and comparing results for different domains, Cronbach's alpha coefficients range from 0.92 to 0.94.

General Heath Questionnaires GHQ-28 by David Goldberg, adapted into the Polish by Makowska and Merecz, the reliability of the test ranges from +0.82 to 0.93.

Also, could you clarify how questionnaires were distributed? Were they mailed to participants or handed out in person?

Response: Changed to: The inclusion criterion was age 40 years old or above and work as a nurse. Nurses working in hospital, various types of acute ward, primary health care, outpatient specialist care and other who agreed to participate were included in the study. The questionnaires were given in a sealed envelope to all the nurses who were 40 years old and older in a selected hospitals. A total of 560 questionnaires were distributed with the collaboration of the Chamber of Nurses and Midwives and in return 523 envelopes were collected and they were the basis for empirical analysis.

Finally, I believe it would be helpful to the reader if you could expand your discussion on recommendations, particularly as they relate to practice. For example, are there resources that could be made available to nurses in the workplace to help them with mental health issues, financial concerns, etc.?

Response: Changed to: It is recommended that this study introduces more psychological support into the practice of nurses, attention to the stressful work environment and the risk of burnout syndrome, as well as more support from the management in financial matters.

Thank you for your useful recommendation and support which will affect the quality of the manuscript.

Round 2

Reviewer 1 Report

Dear Authors,
I think that the paper has improved.
Congratulations.

Author Response

Thank you for the reviews of our paperwork and for your suport.

 English language and style are fine/minor spell check required 

Response: We use a professional MDPI English editing service. We also corrected discussion, methodology and reference section.

Reviewer 2 Report

I appreciate the efforts made by the authors to revise the paper, and for providing the corrected citations which are now easier to access. 

I believe the study will be of interest to many readers, and is a valuable contribution in the topic of wellness in healthcare providers. 

Author Response

Thank you for the reviews of our paperwork and for your suport.

We use a professional MDPI English editing service. We also corrected discussion, methodology and reference section.

Reviewer 3 Report

Comments to Authors

Thank you to the authors for their substantial edits to this manuscript and for adding more relevant literature to your introduction.

Although the authors have added some information to the methodology section, the way the study was conducted as well as the information regarding the use of the questionnaire are unclear. Please provide information whether the authors have the approval for the use of the WHOQOL questionnaire and the identification number providing this.

The manuscript would benefit from a more concise and focused discussion section.

Author Response

Thank you for the reviews of our paperwork and for your suport.

English language and style are fine/minor spell check required 

Response: We use a professional MDPI English editing service.

Although the authors have added some information to the methodology section, the way the study was conducted as well as the information regarding the use of the questionnaire are unclear. Please provide information whether the authors have the approval for the use of the WHOQOL questionnaire and the identification number providing this.

Response:

Polish version of WHOQOL-100 and WHOQOL-Bref have been prepared according to standing standards and the proces was described in detail in the cultural adaptation of a research tool. Polish version WHOQOL were published by book : Wołowicka, L.; Jaracz, K. Polska wersja WHOQOL – WHOQOL 100 i WHOQOL-BREF. [in:] Wołowicka L. (Ed): Jakość życia w naukach medycznych; Akademia Medyczna im. Karola. Marcinkowskiego, Poznań 2001, pp 235-289.

and it is made available for researchers respecting the copyright.

 and an article:

Jaracz K, Kalfoss M, Górna K, Baczyk G. Quality of life in Polish respondents: psychometric properties of the Polish WHOQOL-Bref. Scand J Caring Sci. 2006 Sep;20(3):251-60. PubMed PMID: 16922978.

We have completed the methodological section with data on validity of the instrument.

The authors (Prof. K. Jaracz) have allowedus to use this tool non-commercially, provided that we properly calculate and interpret the results and refer these publications.

 The manuscript would benefit from a more concise and focused discussion section.

Response:

Corrected discussion section. We also corrected methodology and reference section.

Reviewer 4 Report

Thank you for addressing my concerns regarding reliability of instrumentation and recommendations for practice. Are there any data on validity of the instruments. Can you be a bit more specific on the recommendations for practice?

Author Response

Thank you for the reviews of our paperwork and for your suport.

Extensive editing of English language and style required 

Response: We use a professional MDPI English editing service.

Thank you for addressing my concerns regarding reliability of instrumentation and recommendations for practice. Are there any data on validity of the instruments.

Response:

We have completed the methodological section with data on validity of the instruments

Jaracz K, Kalfoss M, Górna K, Baczyk G. Quality of life in Polish respondents: psychometric properties of the Polish WHOQOL-Bref. Scand J Caring Sci. 2006 Sep;20(3):251-60. PubMed PMID: 16922978.

Makowska Z, Merecz D, Mościcka A, Kolasa W. The validity of general health

questionnaires, GHQ-12 and GHQ-28, in mental health studies of working people. Int J Occup Med Environ Health. 2002;15(4):353-62.

Can you be a bit more specific on the recommendations for practice?

Response:

Corrected

“It is recommended that more psychological support for nurses is introduced, drawing attention to the stressful work environment and the risk of burnout syndrome, which can significantly affect the care and safety of patients. Also, greater support from the management in terms of financial matters and increased employment is needed, as these factors will translate into improved quality of care and satisfaction of the nurses as well as preventing nurses from leaving the profession.”
